# Persistent deleterious effects of a deleterious *Wolbachia* infection

**Perran A. Ross***[☉], **Jason K. Axford**[☉], **Ashley G. Callahan**[☉], **Kelly M. Richardson, Ary A. Hoffmann**

Pest and Environmental Adaptation Research Group, Bio21 Institute and the School of BioSciences, The University of Melbourne, Parkville, Victoria, Australia

☉ These authors contributed equally to this work.
* perran.ross@unimelb.edu.au

**Data Availability Statement:** All relevant data are within the manuscript and its Supporting Information files.

**Funding:** This work was supported by the National Health and Medical Research Council (1132412

## Abstract

*Wolbachia* are being used to reduce dengue transmission by *Aedes aegypti* mosquitoes around the world. To date releases have mostly involved *Wolbachia* strains with limited fitness effects but strains with larger fitness costs could be used to suppress mosquito populations. However, such infections are expected to evolve towards decreased deleterious effects. Here we investigate potential evolutionary changes in the *w*MelPop infection transferred from *Drosophila melanogaster* to *Aedes aegypti* more than ten years (~120 generations) ago. We show that most deleterious effects of this infection have persisted despite strong selection to ameliorate them. The *w*MelPop-PGYP infection is difficult to maintain in laboratory colonies, likely due to the persistent deleterious effects coupled with occasional maternal transmission leakage. Furthermore, female mosquitoes can be scored incorrectly as infected due to transmission of *Wolbachia* through mating. Infection loss in colonies was not associated with evolutionary changes in the nuclear background. These findings suggest that *Wolbachia* transinfections with deleterious effects may have stable phenotypes which could ensure their long-term effectiveness if released in natural populations to reduce population size.

## Author summary

Mosquitoes infected with *Wolbachia* bacteria are being deployed into the field where they can suppress mosquito populations and reduce dengue transmission. These programs rely on the use of *Wolbachia* strains that have desirable phenotypes, which can include deleterious fitness effects, reproductive manipulation and virus blocking. However, theory predicts that *Wolbachia* will evolve to become less costly to their hosts, reducing the effectiveness of these programs. We investigate the potential for evolutionary changes by performing a comprehensive phenotypic analysis of a deleterious *Wolbachia* strain, *w*MelPop-PGYP, that was introduced to *Aedes aegypti* mosquitoes from *Drosophila* over ten years ago. In contrast to theoretical expectations and research from *Drosophila*, our results suggest that *Wolbachia* strains with deleterious effects may have stable phenotypes, ensuring their long-term effectiveness if released into natural populations.

and 1118640 to AAH, www.nhmrc.gov.au) and the Australian Research Council (DP190101877 to AAH, www.arc.gov.au). The funders had no role in study design, data collection and analysis, decision to publish, or preparation of the manuscript.

**Competing interests:** The authors have declared that no competing interests exist.

## Introduction

There is increasing interest in using *Wolbachia* bacterial infections for suppressing dengue transmission by mosquitoes, with field releases aimed at both replacing existing natural mosquito populations with those infected by *Wolbachia* [1, 2] and suppressing these populations through sterility induced by *Wolbachia*-infected males [3]. Replacement releases can be effective because the presence of *Wolbachia* in mosquitoes reduces transmission of arboviruses [4–6]. In addition, *Wolbachia* decreases the fitness of its mosquito hosts [7]. While this might have a suppressive effect on dengue transmission, for instance, by shortening mosquito lifespan [8], it can make the infections more difficult to introduce into populations because the initial *Wolbachia* frequency must be higher for the population to be invaded by *Wolbachia* [9].

The *w*MelPop infection, which originated from a laboratory strain of *Drosophila melanogaster*, was one of the first *Wolbachia* strains successfully introduced into *Aedes aegypti* [10] where it is very effective at blocking transmission of dengue and other arboviruses [5]. *w*MelPop in *Ae. aegypti* represents a variant referred to as *w*MelPop-PGYP which lacks the Octomom genomic region present in the original strain [11]. The *w*MelPop strain reduces longevity in *D. melanogaster* [12] while *w*MelPop-PGYP in mosquitoes has additional deleterious effects, including reduced viability of eggs maintained in a quiescent state [13, 14]. The *w*MelPop-PGYP infection was released in field trials in Vietnam and Australia but failed to establish [15], although it successfully invaded semi-field cages [6]. Because of these deleterious effects, *w*MelPop may represent an effective tool to reduce or even eliminate mosquito populations [16], particularly in isolated populations experiencing seasonal rainfall [17].

One of the challenges in using *w*MelPop-PGYP is that the strain can be difficult to maintain under laboratory conditions, with the infection occasionally being lost from colonies. For instance, on one occasion we found that 95.5% (43/45) of our colony was infected based on RT-PCR screening but this declined to 6.7% (2/30) four months later. Although the infection causes strong cytoplasmic incompatibility and shows near-complete maternal transmission, which allow *Wolbachia* infections to invade populations once an unstable equilibrium frequency dictated by deleterious fitness effects is exceeded [6], the infection may still be lost for unknown reasons even when it is detected at a high frequency with molecular assays. Environmental effects might reduce infection frequencies since high temperatures and low levels of antibiotics can clear *Wolbachia* infections [18, 19]. However, there is normally careful control of temperature and antibiotics in laboratory cultures. Other factors that may contribute to infection loss are inappropriate storage of eggs coupled with sporadic incomplete maternal transmission.

While *Wolbachia* infections like *w*MelPop and *w*Au [4] reduce host fitness, their effects are expected to attenuate over time because any *Wolbachia* or host alleles that decrease deleterious fitness effects should be favoured by selection [9, 20]. Evidence for such a process has been obtained for the *w*Ri infection of *Drosophila simulans* where an initially deleterious effect on offspring production has attenuated to the extent that *w*Ri infected *D. simulans* now have a higher production rate than uninfected females [21]. This could undermine any strategy that relies on maintaining deleterious fitness effects after *Wolbachia* are established in novel hosts, a process that has been documented for *w*MelPop after transfer to *D. simulans* [22, 23]. Evolutionary changes in the nuclear background may also suppress the phenotypic effects of *Wolbachia*, as demonstrated by the evolution of male-killing suppression in butterflies [24]. Although *w*MelPop continues to impose deleterious effects in its native host *D. melanogaster* after many years of laboratory culture [25], it is unclear if deleterious effects and the ability to cause cytoplasmic incompatibility have persisted in the derived *w*MelPop-PGYP infection of *Ae. aegypti*.

To investigate these issues, we consider whether there have been evolutionary changes in *w*MelPop-PGYP or its *Ae*. *aegypti* host in the 10-year period since the infection was established by comparing recent and past data on phenotypic effects of the infection. We also investigate factors that may confound monitoring of *w*MelPop-PGYP and contribute to instability of the infection in laboratory cultures.

## Methods

### Ethics statement

Blood feeding of female mosquitoes on human volunteers for this research was approved by the University of Melbourne Human Ethics Committee (approval 0723847). All adult subjects provided informed written consent (no children were involved).

### Mosquito strains and colony maintenance

We performed experiments with our laboratory populations of *w*MelPop-PGYP-infected [10], *w*Mel-infected [6], *w*AlbB-infected [26] and uninfected *Ae*. *aegypti* mosquitoes. The *w*Mel-Pop-PGYP transinfection in *Ae*. *aegypti* (which we hereafter refer to simply as *w*MelPop except where clarification is required) was derived from *D*. *melanogaster* [12] and was passaged in a mosquito cell line before being introduced into *Ae*. *aegypti* through embryonic microinjection [10]. *w*MelPop-infected mosquitoes were collected from Babinda, Queensland, Australia in 2012, three months after releases commenced [14] and maintained in the laboratory since collection. All *Wolbachia*-infected populations were backcrossed to a common Australian nuclear background for at least five generations to ensure that backgrounds were >98% similar [14]. Stock populations were maintained through continued backcrossing to uninfected North Queensland material every six generations. Mosquitoes were reared in a temperature-controlled laboratory environment at 26˚C ± 1˚C with a 12 hr photoperiod according to methods described previously [27, 28]. Larvae were reared in trays filled with 4 L of reverse osmosis water at a controlled density of 450 larvae per tray. Larvae were fed TetraMin tropical fish food tablets (Tetra, Melle, Germany) *ad libitum* until pupation. Female mosquitoes from all laboratory colonies and experiments were blood fed on the forearms of human volunteers. For colony maintenance, females were blood fed approximately one week after adult emergence, with eggs normally hatched within one week of collection. Only eggs from the first gonotrophic cycle were used to establish the next generation. An uninfected population (denoted *w*Mel-Pop-negative) was derived from *w*MelPop females that had lost their *Wolbachia* infection in June 2019. The *w*MelPop-negative population was used in life history experiments and to test for nuclear background evolution. All experiments were performed in 2019 except for the first *Wolbachia* mating transmission experiment (performed in 2016) and the routine scoring of egg hatch from 2012–2018.

### *Wolbachia* screening

*Aedes aegypti* females were tested for the presence of *Wolbachia* DNA using methods previously described with modifications [27, 29]. DNA extraction methods varied between experiments due to our research spanning seven years. Mosquito DNA was extracted using 100–250 μL of 5% Chelex solution (Bio-Rad Laboratories, Gladesville, NSW, Australia) and 2.5–5 μL of Proteinase K (20 mg/mL, Bioline Australia Pty Ltd, Alexandria, NSW, Australia) in either 96-well plates or 1.5 mL tubes. Polymerase chain reactions were carried out with a Roche LightCycler 480 system (384-well format, Roche Applied Science, Indianapolis, IN, USA) using a RT/HRM (real-time PCR/high-resolution melt) assay as described previously [27, 29].

We used mosquito-specific (*mRpS6*), *Aedes aegypti*-specific (*aRpS6*) and *Wolbachia*-specific primers (*w1* primers for the *w*MelPop and *w*Mel infections and *w*AlbB primers for the *w*AlbB infection) to diagnose *Wolbachia* infections [27](S1 Table). All individuals were expected to have robust and similar amplification of the *mRpS6* and *aRpS6* primers. An individual was scored as positive for *Wolbachia* if its *w1* or *w*AlbB Cp (crossing point) value was lower than 35 and its Tm (melting temperature) value was within the expected range based on positive controls (approximately 84.3, but this varied between runs). An individual was negative for *Wolbachia* when Cp values were 35 or absent and/or Tm values were inconsistent with the controls. For experiments with the *w*MelPop infection, we assigned infected individuals to two categories: strongly positive (Cp $\leq$ 23) and weakly positive (Cp > 23). Based on the mating transmission experiments (see below), females that were strongly positive likely represented true infections, while weakly positive females were likely uninfected and had mated with a *Wolbachia*-infected male. Relative *Wolbachia* densities were determined by subtracting the *Wolbachia* Cp from the *aRpS6* Cp and then transforming this value by $2^{n}$.

## Re-evaluation of deleterious effects

The *w*MelPop-PGYP infection induces a range of deleterious effects, including life shortening, reduced fertility, impaired blood feeding success and reduced quiescent egg viability as outlined below. We re-evaluated these deleterious effects by performing experiments with the *w*MelPop infection over 10 years after its introduction to *Ae. aegypti*. Before experiments commenced, the *w*MelPop-infected colony was purified by pooling the offspring of isolated females that were strongly positive for *Wolbachia* (see *Infection recovery*). Female offspring were crossed to uninfected males, and the progeny were used in the following experiments. We compared fitness relative to two uninfected populations; a natively uninfected laboratory population (uninfected) and a population derived from uninfected individuals from the *w*Mel-Pop colony that had lost their infection (*w*MelPop-negative). Due to logistical constraints, the fertility experiment included the *w*MelPop and *w*MelPop-negative populations only.

**Longevity.** Previous studies reported that *w*MelPop shortens adult lifespan by approximately 50% [10, 14]. We performed longevity assays by establishing 8 replicate 3 L cages with 50 adults (25 males and 25 females) for each population. Cages were provided with 10% sucrose and water cups which were replaced weekly. Females were provided with blood meals for 10 minutes once per week and given constant access to an oviposition substrate. Mortality was scored three times per week by removing and counting dead adults from each cage until all adults had died. One replicate of *w*MelPop was discarded due to a sugar spill early in the experiment which caused high mortality. We used log-rank tests to compare adult longevity between populations. To evaluate *Wolbachia* density and infection frequencies with adult age, 16 females from separate cages that were 0, 7, 14, 21, 28 and 35 d old were screened for *Wolbachia*. We used a linear regression to test whether (log) *Wolbachia* density was affected by adult age. All data were analyzed using SPSS statistics version 24.0 for Windows (SPSS Inc, Chicago, IL).

**Fertility.** The *w*MelPop-PGYP infection substantially reduces fertility as females age [13]; we therefore tested the fertility of *w*MelPop and *w*MelPop-negative populations over successive gonotrophic cycles. The uninfected population was not included in this experiment. We established two cages of approximately 500 individuals (equal sex ratio) for each population. Five-day old females (starved for 1 d) were blood fed on the forearm of a human volunteer. Thirty-five engorged females were selected randomly from each population and isolated in 70 mL cups with sandpaper strips and larval rearing water to encourage oviposition. Eggs were collected 4 days after blood feeding, partially dried and hatched three days after collection.

Fecundity and egg hatch proportions were determined by counting the number of unhatched and hatched eggs (hatched eggs having a clearly detached cap). Following egg collection, females were returned to their respective cages for blood feeding. Successive gonotrophic cycles were initiated every 4–5 days with females selected randomly from cages. Cages were provided with oviposition substrates, however no sugar was provided to isolated females or the population cage during the experiment because sugar feeding influences fecundity [30]. We tested fertility for a total of 9 gonotrophic cycles. Females from the wMelPop population that were still alive after 9 gonotrophic cycles were tested with qPCR to confirm Wolbachia infection. Effects of gonotrophic cycle on egg hatch proportions were compared for the wMelPop and wMelPop-negative populations. Egg hatch proportions were not normally distributed and were therefore analysed with Kruskal-Wallis tests.

**Quiescent egg viability.** The wMelPop infection reduces the viability of quiescent eggs [13, 14, 16]. For quiescent egg viability assays, eggs were collected from colonies on sandpaper strips and stored in a sealed container with a saturated solution of potassium chloride to maintain ~80% humidity. Nine replicate batches of eggs (40–98 eggs per batch) per population were hatched twice per week by submerging eggs in containers of water with a few grains of yeast. Egg hatch proportions were determined by dividing the number of hatched eggs by the total number of eggs. Larvae that had not completely eclosed and died in the egg were scored as unhatched. This experiment continued until eggs were 31 d old. Effects of egg storage duration on hatch proportions were compared for the wMelPop, wMelPop-negative and uninfected populations Egg hatch proportions were not normally distributed and were therefore analysed with Kruskal-Wallis tests. To test for the potential loss of wMelPop infection with egg storage, we reared larvae hatching from 3, 13, 20, 24, 27 and 31 d old egg to adulthood and scored 16 females (< 24 hr old) for Wolbachia infection and density from each group. We used a linear regression to test whether (log) Wolbachia density was affected by egg storage duration.

**Blood feeding success.** The wMelPop infection reduces female blood feeding success and affects probing behaviour, particularly in older females [31, 32]. We evaluated blood feeding traits in 5 and 35 d old females according to methods described previously [33]. We recorded pre-probing duration (time from landing to insertion of the proboscis), feeding duration, blood meal weight and proportion feeding. Females that did not feed within 10 minutes were scored as not feeding. The proportion of females exhibiting a bendy or shaky proboscis phenotype [31, 32] was also recorded. Feeding trials were performed on individual females by three experimenters. At least 32 individuals per population and age group were tested across the three experimenters. To confirm the infection status of wMelPop females, we screened all 35 d old females for Wolbachia infection. Pre-probing duration, feeding duration and blood meal weight data were analysed with general linear models, with population (wMelPop, wMelPop-negative and uninfected) and experimenter (the person being fed on by the mosquito) included as factors. Pre-probing and feeding durations were log transformed for normality before analysis. Comparisons of proportional data (proportion feeding and the presence of a bendy or shaky proboscis) with previous studies were performed with two proportions Z-tests.

## Loss of *Wolbachia* during colony maintenance

We carried out a series of experiments and monitoring exercises to understand the loss of the wMelPop infection in colonies during routine maintenance.

**Infection recovery.** In May 2019 we observed an apparent loss of wMelPop infection from our laboratory colony despite a high level of infection in previous generations. To return the population to a 100% infection frequency, one hundred blood-fed females were isolated

for oviposition, screened for *Wolbachia*, then placed into categories of strongly positive, weakly positive or negative (see *Wolbachia* screening). We then pooled the offspring of females from each category and screened 30 offspring (15 males and 15 females) for *Wolbachia* per category. Female offspring from the strongly positive population were crossed to uninfected males before commencing the maternal transmission, nuclear background evolution and life history experiments.

**Maternal transmission.**   We estimated maternal transmission fidelity by crossing *w*Mel-Pop-infected females to uninfected males, then screening ten offspring (4th instar larvae) from the first gonotrophic cycle of ten females that had been separated individually for oviposition. Maternal transmission fidelity was expressed as the proportion of infected offspring produced by infected mothers, for which 95% binomial confidence intervals were calculated.

**Nuclear background evolution.**   Loss of *w*MelPop infection in laboratory colonies may be explained by the evolution of resistance to *Wolbachia* infection by uninfected mosquitoes. We performed crossing experiments to test whether the *w*MelPop infection was maintained across generations when *w*MelPop-infected females were crossed to natively uninfected males or uninfected males that had lost their *Wolbachia* infection (*w*MelPop-negative). We established two replicate populations for each cross with 200 adults of each sex. Males and females were separated as pupae and then crossed when adults were 3–5 d old. Crosses were performed for four consecutive generations, with each cage maintained according to our regular colony maintenance schedule (females were blood fed approximately one week after emergence and eggs hatched within one week of collection). Thirty individuals from each replicate population per generation were then screened for *Wolbachia* infection. A *w*MelPop colony (*w*MelPop-infected males crossed with *w*MelPop-infected females) was also monitored across the same time period.

To test for resistance to cytoplasmic incompatibility, we tested the ability of *w*MelPop-infected males to induce cytoplasmic incompatibility with uninfected and *w*MelPop-negative females. For each cross, 30 males and 30 females were aspirated into a single 3 L cage. When adults were 5 d old, females were blood fed. Twenty females from each cross were isolated for oviposition and egg hatch proportions were determined according to the fertility experiment (see above).

**Wolbachia mating transmission.**   Although *Wolbachia* in mosquitoes are maternally transmitted, it is possible that *Wolbachia* might also be transferred through seminal fluid, leading to the detection of *Wolbachia* in uninfected females that mate with infected males. To test for *Wolbachia* transmission through mating, we performed crosses between *Wolbachia*-infected males and uninfected females. Experiments were performed with the *w*MelPop, *w*Mel and *w*AlbB strains. Control crosses were also performed, where both sexes were either infected (positive controls) or uninfected (negative control). Crosses were established with 160 virgin adults of each sex (4–7 d old) in a single cage and left for two days to mate, after which males were removed. Females were blood-fed one week after crosses were established and provided with an oviposition substrate. Thirty females (whole adults) were stored 2, 9, 16 and 23 d after crosses were established and screened for *Wolbachia*. Females from the positive and negative controls were tested 2 and 23 d after crosses were established. Due to apparent differences in mating transfer between *Wolbachia* strains, this experiment was repeated with the *w*AlbB infection, but females were stored 5 d after crosses were established.

We conducted an additional cross between uninfected females and *w*MelPop-infected males to see if the detection of *Wolbachia* following transmission through mating was tissue-specific. Females and males were left to mate for five days, after which females were stored in ethanol. Heads and abdomens from 20 uninfected females were dissected and extracted separately for *Wolbachia* screening.

**Relative fitness during laboratory maintenance.** We compiled data on egg hatch proportions during our routine maintenance of *w*MelPop, *w*Mel, *w*AlbB and uninfected colonies from July 2012 to April 2018. Egg hatch proportions were determined by hatching a subset of eggs collected from each colony during maintenance (>200 eggs per subset), then dividing the number of larvae counted by the number of eggs tested. We then divided the egg hatch proportions of *Wolbachia*-infected colonies by the egg hatch proportion of the uninfected colony to obtain relative egg hatch proportions. When multiple *Wolbachia*-infected colonies were maintained simultaneously, we included these as separate estimates. We used sign tests to compare relative hatch proportions of *Wolbachia*-infected and uninfected colony eggs. We used a general linear model to test for long-term changes in the relative egg hatch proportion of *w*Mel-Pop-infected colonies.

## Results

### Re-evaluation of deleterious effects

We re-evaluated the deleterious fitness effects induced by *w*MelPop-PGYP to test for attenuation. In previous experiments conducted more than 10 years ago, the *w*MelPop-PGYP infection shortened adult male and female lifespan by ~50% relative to uninfected populations [10, 14]. Here, the *w*MelPop-PGYP infection shortened median female lifespan by 22% compared to the uninfected populations (Log-rank: $\chi^2$ = 116.310, df = 2, P < 0.001), while male lifespan was unaffected by population ($\chi^2$ = 4.722, df = 2, P = 0.094, Fig 1). These results suggest that the effects of *w*MelPop on adult lifespan may have attenuated, though direct comparisons with previous studies are difficult since experimental conditions will vary. Although adults from this experiment were not screened for *Wolbachia*, samples of colony females from the same generation aged 0–35 d (n = 101) all had strongly positive (Cp ≤ 23) infections, suggesting that this result was not influenced by incomplete maternal transmission. (log) *Wolbachia* density decreased with adult age (linear regression: $R^2$ = 0.186, $F_{1,86}$ = 20.837, P < 0.001, S1A Fig), in contrast to *Drosophila* where *w*MelPop density [34, 35] (and to a lesser extent, *w*MelPop-CLA density [36]) increases with age.

In previous studies, *w*MelPop infection reduced fertility with increasing female age [13] and egg storage duration [13, 14]. In the current experiment, *w*MelPop infection reduced fecundity by 22.54% and egg hatch by 11.44% overall, indicating that deleterious effects have

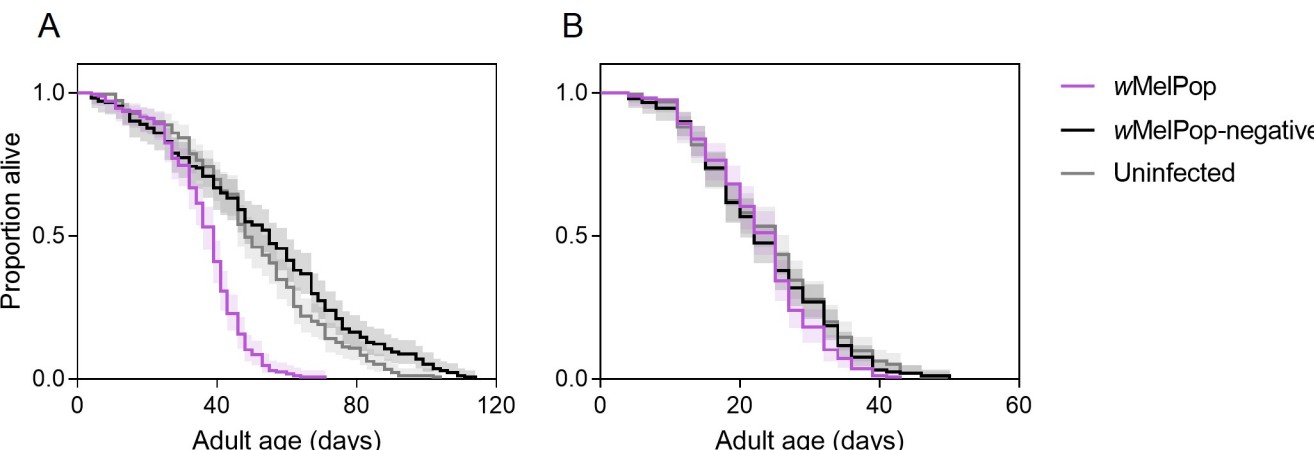

**Fig 1. Longevity of female (A) and male (B) adult *Aedes aegypti* from *w*MelPop (purple lines), *w*MelPop-negative (black lines) and uninfected (gray lines) populations.** Lines represent the proportion of mosquitoes alive, while shaded regions show 95% confidence intervals.

persisted for over 10 years after transinfection. The viability of $w$MelPop-infected eggs declined rapidly with increasing storage duration (Kruskal-Wallis: $\chi^2$ = 69.307, df = 8, P < 0.001, Fig 2D) but hatch proportions for $w$MelPop-negative ($\chi^2$ = 7.199, df = 8, P = 0.515) and uninfected ($\chi^2$ = 5.503, df = 8, P = 0.703) eggs were stable across the same duration. Patterns of fecundity (Fig 2A) and quiescent egg viability (Fig 2D) observed here were similar to a previous study [13] although experimental conditions would have differed somewhat. Loss of female fertility with age was due to declining fecundity rather than egg hatch, which was stable across gonotrophic cycles for both $w$MelPop (Kruskal-Wallis: $\chi^2$ = 4.654, df = 7, P = 0.702) and $w$MelPop-negative ($\chi^2$ = 7.580, df = 8, P = 0.476) females (Fig 2B).

As adult age increased, we observed an increasing proportion of $w$MelPop females that had a high egg production but had zero eggs hatching (Fig 2C). We excluded these individuals from the results since they may represent uninfected mosquitoes that mated with $w$MelPop-infected males. Uninfected individuals may result from incomplete maternal transmission and become increasingly represented throughout the experiment due to having a longer lifespan (Fig 1A). Only two of the seven $w$MelPop females surviving to the ninth gonotrophic cycle had a strongly positive (Cp ≤ 23) *Wolbachia* infection, indicating maternal transmission leakage. In contrast, all individuals hatching from quiescent eggs (storage durations of 3–31 d, n = 96) were strongly positive for *Wolbachia* (Fisher's exact test: P < 0.001), although adult *Wolbachia* density decreased with increasing egg storage duration (linear regression: $R^2$ = 0.108, $F_{1,83}$ = 10.087, P = 0.002, S1B Fig).

The $w$MelPop infection reduces female blood feeding success and affects probing behaviour, particularly in older females [31, 32]. Here we found no effect of population on pre-

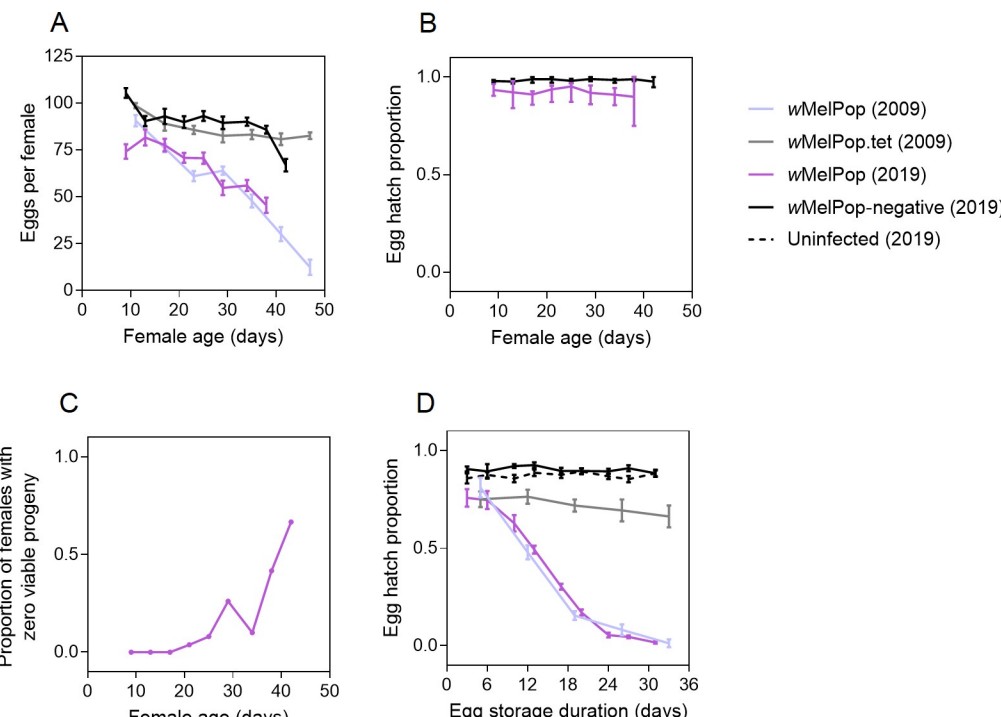

**Fig 2. Fertility of $w$MelPop-infected and uninfected *Aedes aegypti* populations with increasing female age and egg storage duration.** (A) Fecundity across gonotrophic cycles. (B) Egg hatch proportion across gonotrophic cycles. (C) Proportion of $w$MelPop-infected females with zero viable progeny across gonotrophic cycles. (D) Egg hatch proportion with different durations of egg storage. Data for 2009 (pale lines) were manually extracted from McMeniman and O'Neill [13] using ScanIt software (https://www.amsterchem.com/scanit.html). Lines and error bars are means and standard errors respectively, consistent with the original study.

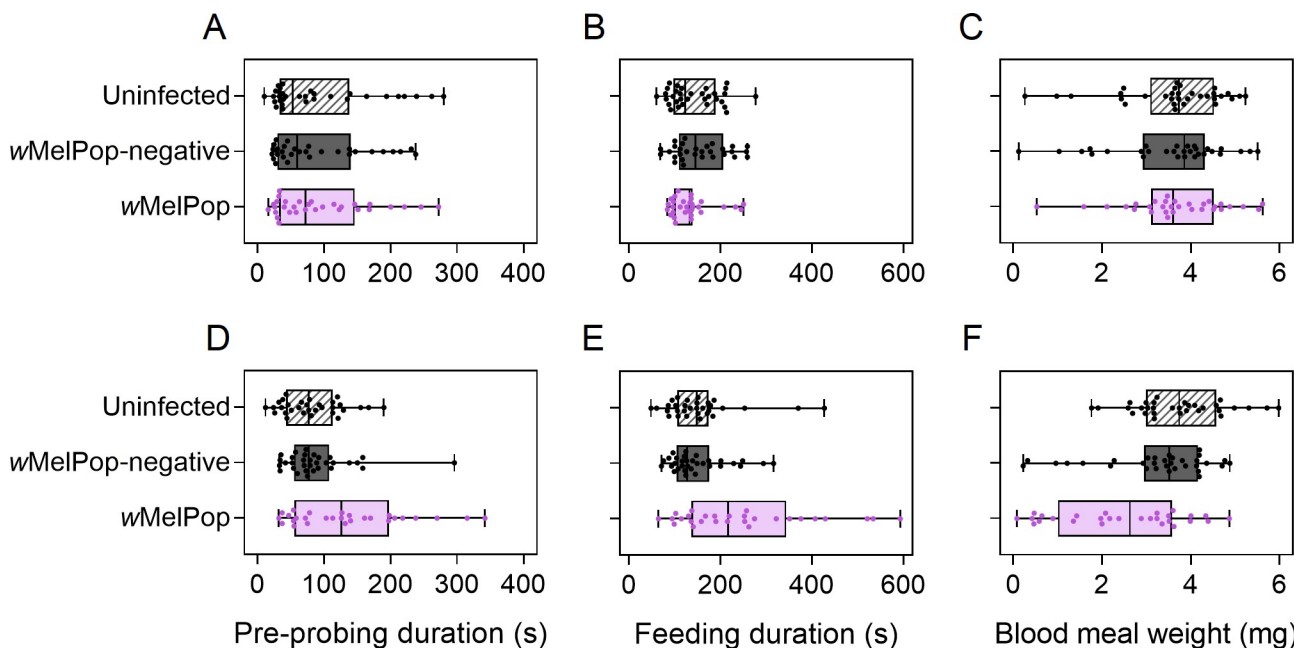

**Fig 3.** Pre-probing duration (A,D), feeding duration (B,E) and blood meal weight (C,F) of uninfected, wMelPop-negative and wMelPop *Aedes aegypti* females aged 5 (A-C) or 35 d (D-F). Box plots show medians and interquartile ranges, with error bars representing minimum and maximum values. Data for individual females are shown by dots.

probing and feeding duration or blood meal weight in 5 d old females (GLM: all P > 0.05, Fig 3). Conversely, in 35 d old females we observed costs of wMelPop infection for all traits, with significant effects of population for pre-probing duration ($F_{2,82}$ = 26.135, P < 0.001), feeding duration ($F_{2,82}$ = 7.988, P = 0.001) and blood meal weight ($F_{2,82}$ = 14.338, P < 0.001, Fig 3). Substantial effects of experimenter were also observed for all three traits tested (all P < 0.01), leading to differences of up to 0.37 mg (10.27%) in blood meal weight, 39.5 s (27.96%) in feeding duration and 100 s (113.64%) in pre-probing duration.

Effects of wMelPop infection on blood feeding traits may have been weaker in comparison to previous studies with similar methods. For instance, Turley et al. [31] observed a 50.3% (95% confidence interval: 37.5–63.1%) reduction in blood meal weight in 35 d old females due to wMelPop infection, while we observed a 29.5% (95% confidence interval: 12.1–46.7%) reduction relative to the two uninfected populations. Aged wMelPop females had reduced feeding success (65% feeding compared to 91% for uninfected populations) and also displayed a bendy/shaky proboscis phenotype as characterized previously [31, 32]. However, these phenotypes occurred at a significantly lower frequency than previously reported [32] (proportion feeding: two proportions Z-test: Z = 3.431, P < 0.001, bendy/shaky proboscis: Z = 4.288, P < 0.001). Weaker effects relative to previous studies may result from methodological differences, human experimenter effects, effects of laboratory rearing, attenuation or incomplete maternal transmission. *Wolbachia* screening of 35 d old females showed that 6 females (20%) had a weakly positive (Cp > 23) infection which may indicate maternal transmission leakage.

## Loss of *Wolbachia* during colony maintenance

**Infection recovery.** Due to an apparent loss of *Wolbachia* from our wMelPop-PGYP colony, we isolated females to restore the wMelPop infection in the population. Of the females that produced viable offspring, 20 were negative, 17 were strongly positive (median Cp 16.3,

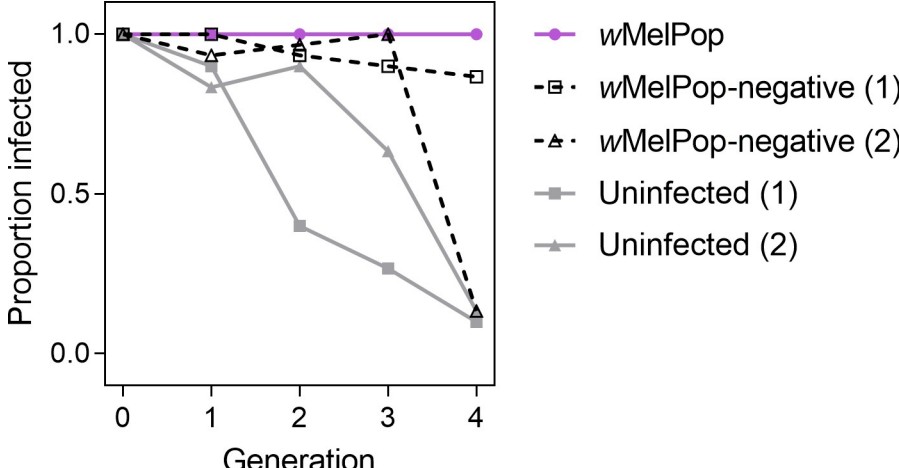

**Fig 4. Loss of *w*MelPop infection in *Aedes aegypti* in the absence of cytoplasmic incompatibility.** *w*MelPop-infected females were crossed to wMelPop-negative (gray), uninfected (gray) or *w*MelPop (purple) males each generation for four generations. Infection frequencies were determined for 30 individuals per population, per generation.

range 3.33) and 41 were weakly positive (median Cp 31.38, range 8.37). These results point to a polymorphic colony despite the colony having been scored as 100% infected prior to this time (all Cp values ≤ 23). All offspring tested from strongly positive females were strongly positive (females: median Cp 19.19, range 0.61), males: median Cp 19.12, range 5.83). No offspring from the weakly positive or negative females were infected (n = 30 each), thus females scored as weakly positive were unable to transmit *w*MelPop to the next generation.

**Maternal transmission.** We tested ten offspring from ten *w*MelPop-infected females and found that a single female produced two uninfected offspring, with an overall maternal transmission fidelity of 98% (binomial confidence interval: 92.96–99.76%). These results are consistent with previous studies that indicate a low level of maternal transmission failure [14, 18].

**Nuclear background evolution.** We crossed *w*MelPop-infected females to *w*MelPop-negative or uninfected males for four generations to see if the loss of *w*MelPop infection was associated with changes in the nuclear background. The *w*MelPop infection frequency declined in all four populations (Fig 4). In contrast, when *w*MelPop-infected females were crossed to *w*MelPop-infected males the infection frequency remained at 100%, likely due to cytoplasmic incompatibility. Loss of *w*MelPop infection does not appear to be strongly related to nuclear background since the infection declined in both sets of crosses. Rather, declines in infection frequency are likely due to a combination of incomplete maternal transmission and fitness costs.

*w*MelPop-infected males induced complete cytoplasmic incompatibility with uninfected females (no eggs hatching, Table 1), suggesting that this phenotype has remained stable since transinfection over 10 years ago [10]. Compatible crosses exhibited high hatch proportions, showing that the *w*MelPop infection is self-compatible. *w*MelPop-infected males also induced complete cytoplasmic incompatibility with *w*MelPop-negative females, indicating that this population has not evolved resistance to cytoplasmic incompatibility.

**Wolbachia mating transmission.** We crossed *Wolbachia*-infected males with uninfected females to test the potential for *Wolbachia* to be transferred through mating. In control crosses, *Wolbachia*-infected females had a 100% infection frequency and high densities (Fig 5), while *Wolbachia* were not detected when uninfected females were crossed to uninfected males.

**Table 1. Egg hatch proportions resulting from crosses between *w*MelPop, *w*MelPop-negative and uninfected *Aedes aegypti* populations.**

| | | Male | | |
|---|---|---|---|---|
| | | *w*MelPop | Uninfected | *w*MelPop-negative |
| Female | *w*MelPop | 0.933 (0.903, 0.964) | 0.988 (0.970, 1) | Not tested |
| | Uninfected | 0 (0, 0) | 0.936 (0.893, 0.969) | Not tested |
| | *w*MelPop-negative | 0 (0, 0) | Not tested | 0.980 (0.972, 0.984) |

Data are medians followed by 95% confidence intervals (lower, upper).

We detected *Wolbachia* in uninfected females that were crossed to *w*MelPop- (Fig 5A) and *w*Mel-infected (Fig 5B) males for up to 23 d post-mating, with the proportion scored as positive decreasing with time after mating. *Wolbachia* densities in uninfected females were distinctly lower than in females with a maternally-inherited *Wolbachia* infection. In an additional cross, we specifically tested for transfer of seminal fluid by crossing uninfected females to *w*MelPop-infected males and testing the heads and abdomens of females separately. All heads were negative for *Wolbachia*, while 19/20 abdomens were positive with a median Cp of 28.78 (range 4.44). Uninfected females can therefore be incorrectly scored as infected if they have mated with a *w*MelPop or *w*Mel-infected male.

In contrast to the other two infections, we did not detect *Wolbachia* in any uninfected females that were crossed to *w*AlbB-infected males (Fig 4C). We detected no *Wolbachia* in a second independent experiment, indicating that this *Wolbachia* strain is not transferred through mating. Furthermore, we found no evidence for *Wolbachia* transfer through mating in two *Drosophila* species, even for the *w*Mel infection in *D. melanogaster* (S1 Appendix).

**Relative fitness during laboratory maintenance.** We monitored egg hatch proportions of our *Wolbachia*-infected laboratory colonies across multiple generations to assess variance in fitness costs. *w*MelPop-infected (Sign test: Z = 6.197, P < 0.001) and *w*Mel-infected (Z = 3.900, P < 0.001) colonies tended to have lower egg hatch proportions relative to uninfected colonies (Fig 6). *w*AlbB-infected colonies had similar hatch proportions to uninfected colonies overall (Z = 1.000, P = 0.317), though the sample size for this infection was much lower. For the *w*Mel-Pop infection, relative egg hatch proportions were as low as 40% which may contribute to the loss of infection from colonies. Because data were collected over nearly a 6-year period, we could test for changes in egg hatch across time. For *w*MelPop, where the most data were available, there was no temporal difference in relative egg hatch (General linear model: $F_{17,42}$ = 1.727, P = 0.076), suggesting that there has been no major change in relative fitness during this period. These results are consistent with a compilation of fitness estimates from previous studies showing that *w*MelPop consistently induces fertility costs while effects of other *Wolbachia* infections are weaker (S2 Fig, [7]).

## Discussion

Here we provide data that suggests limited evolutionary attenuation of deleterious effects in *w*MelPop-PGYP cultures, either through changes in the host nuclear genome or the *Wolbachia* genome. This is despite an elapsed period of more than ten years or ~120 generations of rearing in the laboratory (and with an additional short period in the field). This contrasts sharply with the attenuation of *w*MelPop seen in *D. simulans* following its transfer from *D. melanogaster*, although the *w*MelPop-PGYP strain in *Ae. aegypti* differs genomically from the *Drosophila* strain, particularly for the Octomom region associated with *Wolbachia* virulence [25]. As in its native host, *w*MelPop reduced longevity when transferred to *D. simulans* [37], *Ae. aegypti* [10] and *Aedes albopictus* [38]. Other deleterious effects in *D. simulans* were also detected;

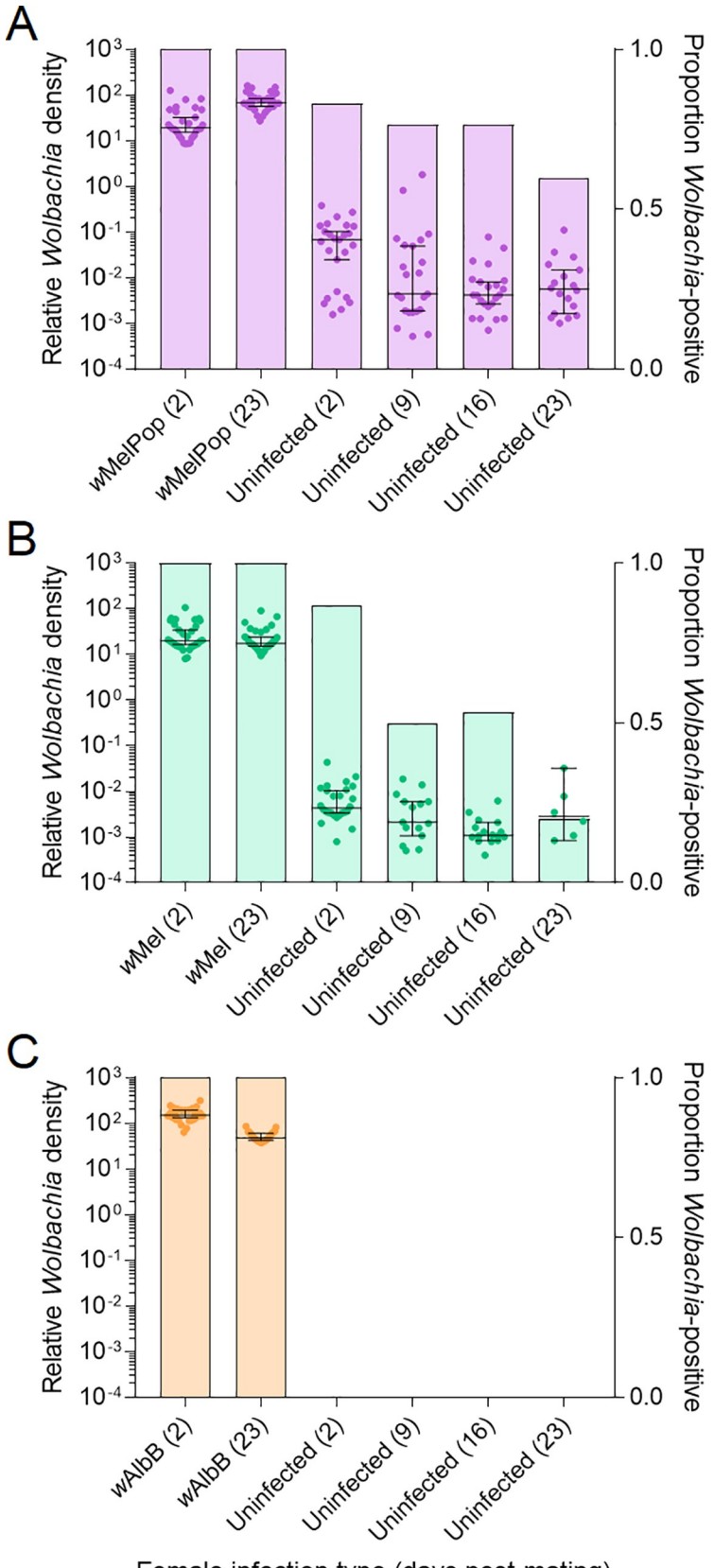

**Fig 5. Detection of *Wolbachia* in uninfected *Aedes aegypti* females via seminal fluid from *Wolbachia*-infected males.** Males were infected with the (A) *w*MelPop, (B) *w*Mel or (C) *w*AlbB *Wolbachia* strains. Dots show *Wolbachia* densities of individual females (left y-axis), while horizontal lines and error bars are medians and 95% confidence intervals respectively. Shaded bars show proportions of females (n = 30) from each group that tested positive for *Wolbachia* (right y-axis).

however, many of these attenuated after around 20 generations, including effects on egg hatch [34]. Moreover, after around 200 generations, *w*MelPop-infected *D. simulans* lines no longer showed a decrease in longevity in some genetic backgrounds [22].

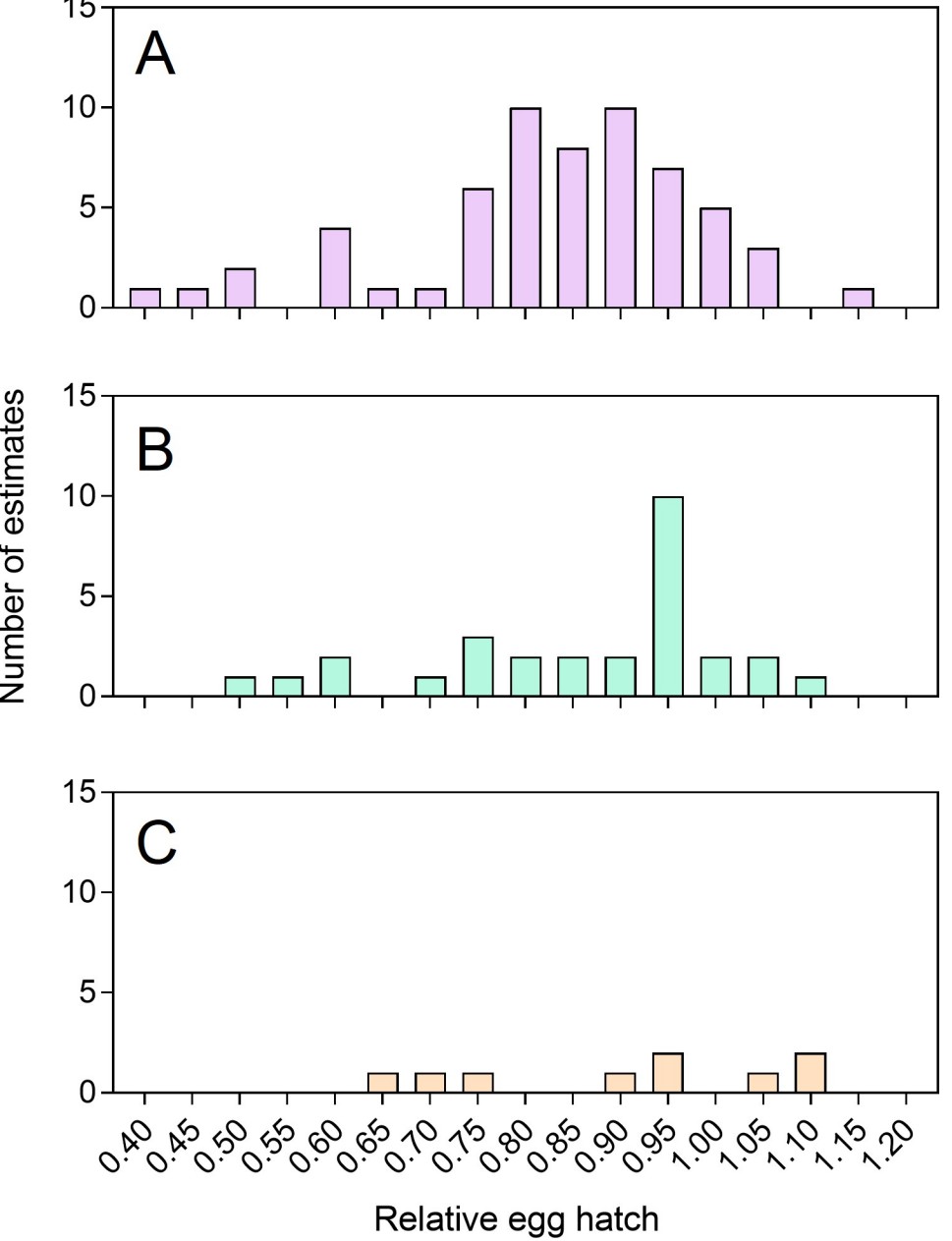

**Fig 6. Histograms of egg hatch proportions of (A) *w*MelPop, (B) *w*Mel and (C) *w*AlbB colonies relative to uninfected colonies during routine laboratory maintenance.** Each estimate was undertaken on a different laboratory generation or colony from at least 200 eggs.

It is unclear why most deleterious effects in *Ae. aegypti* appear to have persisted. Although our laboratory maintenance schedule should reduce the potential for selection, fitness costs are apparent even under benign conditions (such as during the first gonotrophic cycle in the laboratory). Compared to studies performed over ten years ago, some deleterious effects of *w*MelPop appear weaker, particularly blood feeding traits [31, 32] and male longevity [10, 14]. Although this may indicate attenuation, direct comparisons with previous studies are difficult due to methodological differences and potential confounding effects of inbreeding, drift and laboratory adaptation that can occur during colony maintenance [39]. Our observations could in part be explained by the fact that the *w*MelPop line tested here experienced past selection for attenuation. *w*MelPop went through substantial genetic adaptation to the mosquito cell line [36] with reduced virulence, but then experienced no genomic changes after four years within *Ae. aegypti* mosquitoes [11]. Our line also experienced a brief period in the field, which is likely to have imposed strong selection for attenuation. Selection experiments for increased quiescent egg viability in *w*MelPop-infected *Ae. aegypti* found evidence for attenuation, however this involved nuclear background evolution rather than *Wolbachia* evolution [16].

Because *Wolbachia* are maternally inherited, selection acts to increase maternal transmission fidelity and not the ability of males to induce cytoplasmic incompatibility [20]. Novel *Wolbachia* infections tend to induce much stronger cytoplasmic incompatibility than natural infections, suggesting that these effects can attenuate [40]. Furthermore, theory predicts that resistance to cytoplasmic incompatibility may evolve if maternal transmission is incomplete [41]. Although hosts may evolve resistance to the effects of *Wolbachia* on reproduction, such as male killing in *Hypolimnas bolina* [24] and cytoplasmic incompatibility in *D. melanogaster* [9], effects can also remain stable despite intense selection pressure [42, 43]. Over ten years after *w*MelPop was introduced to *Ae. aegypti*, the infection still induces complete cytoplasmic incompatibility. We therefore find no evidence to suggest that cytoplasmic incompatibility has attenuated or that *Ae. aegypti* has evolved to suppress cytoplasmic incompatibility. In crossing experiments, the *w*MelPop infection was lost from colonies regardless of whether infected females were crossed to uninfected males or males that had lost the *w*MelPop infection, suggesting that loss of *w*MelPop was not due to paternal factors that affect *Wolbachia* maternal transmission.

The persistence of deleterious fitness effects may contribute to the occasional loss of the *w*MelPop-PGYP infection from *Ae. aegypti* laboratory populations. Following Hoffmann *et al.* [44] the change in frequency of the infection ($p_f$) in a population is given by

$$p_{(t+1)} = \frac{p_t(1-u)(1-s_f)}{1 - s_f p_t - s_h p_t(1-p_t) - u s_h p_t^2(1-s_f)}$$

where $u$ is the fraction of uninfected progeny produced by infected females, $s_f$ is the fecundity deficit (representing a combination of the number of eggs laid and that hatch) and $s_h$ is the incompatibility between infected and uninfected strains. In the presence of strong maternal transmission ($u = 0$) the unstable point for invasion versus loss of the infection is given by the ratio of $s_f/s_h$ [9]. This means that if incompatibility is very strong ($s_h$ near 1) as is the case with *w*MelPop, it is normally very unlikely for a deleterious fitness effect to result in a loss of infection in a population.

However, we have observed a low level of maternal transmission failure in our *w*MelPop colony of 2%, with an upper estimate of 7%. When coupled with large deleterious effects, this level of leakage may be sufficient to trigger a loss of the *w*MelPop infection. Based on the variance in egg hatch proportions and costs to fecundity, we estimate that the relative fitness of *w*MelPop-infected mosquitoes compared to uninfected mosquitoes may fall to as low as 28%

during routine maintenance, or even lower if adults are aged or eggs are stored before hatching. This will produce a situation where $p_{(t+1)}$ is less than p, and the infection will continue to drop out unless relative fitness is increased.

Our detection of *Wolbachia* at low densities in uninfected females that had mated with *Wolbachia*-infected males was unexpected, given that *Wolbachia* are absent from mature sperm in other insects [45–47]. However, a recent report in *Hylyphantes graminicola* spiders demonstrated sexual transmission of *Wolbachia*, both from males to females and from females to males [48]. Our results have implications for *Wolbachia* monitoring in laboratory and field populations because uninfected females might be incorrectly scored as infected. Assuming random mating, the incidence of false positive detections is equivalent to the frequency of infected individuals in the population. If a loss in infection occurs, it may not be detected immediately when an infection is monitored only by screening adult females. Although false positive individuals in the laboratory can be identified with quantitative assays, determining infection status based on a threshold *Wolbachia* density may be unreliable under field conditions because environmental conditions can affect *Wolbachia* density [18, 19]. We therefore advise that during laboratory maintenance and field monitoring, infection frequencies are determined by screening immature stages, unmated adults or dissected heads. This issue appears to be specific to certain *Wolbachia* strains given that we found no evidence for the transmission through mating of *w*AlbB.

Our findings have implications for the long-term effectiveness of *Wolbachia* releases and for the maintenance of *w*MelPop stocks in the laboratory. The apparent relative stability of deleterious effects shown here suggests that *w*MelPop-PGYP can suppress populations for a long time once established. However, field trials with this infection suggest that long-term persistence in natural populations is unlikely [15]. *w*MelPop-PGYP is difficult to maintain even under benign laboratory conditions due to a combination of incomplete maternal transmission, deleterious effects due to infection, and monitoring issues (false positive detections due to transmission of *Wolbachia* through mating), but a strict rearing schedule and regular *Wolbachia* screening will help to ensure its persistence in a colony.

Due to its fitness costs, *w*MelPop may be suitable for temporary suppression or elimination of populations rather than population replacement which is now taking place in field populations with the *w*Mel and *w*AlbB strains [1, 49]. Suppression through the release of *w*MelPop was proposed as a way of tackling mosquito incursions in isolated areas [17]; as long as such areas are sufficiently isolated to reduce the likelihood of a subsequent invasion by uninfected mosquitoes, this approach could suppress or eliminate mosquito populations without the extensive use of pesticides. Establishing *w*MelPop in large semi-field cages and then imposing a dry period that required the persistence of quiescent eggs led to population elimination [16]. Due to cytoplasmic incompatibility and the deleterious effects of infection, releases of *w*MelPop-infected males and females into the field could result in population suppression once high infection frequencies are reached. This approach to suppression does not require sex separation unlike strategies that rely on cytoplasmic incompatibility [50] and could be effective even if the infection does not persist in the long-term. Although research has shifted away from this deleterious *Wolbachia* infection, *w*MelPop may still prove to be useful when seasonal population suppression is desirable.

## Supporting information

**S1 Table. Primers used in qPCR.**
(PDF)

**S1 Fig. Relative *Wolbachia* density of *w*MelPop-infected females with increasing (A) adult age or (B) egg storage duration.** Each dot represents the *Wolbachia* density of a single female, while solid lines join the median densities for each time point.
(TIF)

**S2 Fig. Relative fitness of *Wolbachia*-infected *Aedes aegypti* compared to uninfected *Ae. aegypti* for fertility-related traits (fecundity and egg hatch), compiled from previous studies [7].** Relative fitness is expressed in terms of effect sizes (Hedges' g), where values below zero indicate a fitness cost. Each dot represents a single fitness estimate. Box plots show medians and interquartile ranges, with error bars representing minimum and maximum values.
(TIF)

**S1 Appendix. Lack of *Wolbachia* transmission through mating in *Drosophila melanogaster* and *D. pandora*.**
(DOCX)

## Acknowledgments

We thank Chengjun Li, Tiana Rey and Véronique Paris for assistance with experimental work. We also thank Ewa Chrostek and two anonymous reviewers for their constructive feedback on the manuscript.

## Author Contributions

**Conceptualization:** Perran A. Ross, Ary A. Hoffmann.

**Formal analysis:** Perran A. Ross.

**Funding acquisition:** Ary A. Hoffmann.

**Investigation:** Perran A. Ross, Jason K. Axford, Ashley G. Callahan, Kelly M. Richardson.

**Methodology:** Perran A. Ross, Jason K. Axford, Ashley G. Callahan.

**Supervision:** Ary A. Hoffmann.

**Visualization:** Perran A. Ross.

**Writing – original draft:** Perran A. Ross, Ary A. Hoffmann.

**Writing – review & editing:** Perran A. Ross, Jason K. Axford, Ashley G. Callahan, Kelly M. Richardson, Ary A. Hoffmann.

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
