## [Decision Letter · Decision Letter 0]

3 Feb 2020

Dear Dr. Ross,

Thank you very much for submitting your manuscript "Persistent deleterious effects of
an unstable deleterious Wolbachia infection" for consideration at PLOS Neglected
Tropical Diseases. As with all papers reviewed by the journal, your manuscript was
reviewed by members of the editorial board and by several independent reviewers. The
reviewers appreciated the attention to an important topic. Based on the reviews, we
are likely to accept this manuscript for publication, providing that you modify the
manuscript according to the review recommendations. 

Sincerely,

Sassan Asgari

Guest Editor

Robert Reiner

Deputy Editor

Reviewer's Responses to Questions

**Key Review Criteria Required for Acceptance?**

**Methods**

-Are the objectives of the study clearly articulated with a clear testable hypothesis
stated?

-Is the study design appropriate to address the stated objectives?

-Is the population clearly described and appropriate for the hypothesis being
tested?

-Is the sample size sufficient to ensure adequate power to address the hypothesis
being tested?

-Were correct statistical analysis used to support conclusions?

-Are there concerns about ethical or regulatory requirements being met?

Reviewer #1: The methods seem appropriate to reach the conclusions. I have only a
couple of comments concenring the methodology. 

First, the strain described in this paper is not the Drosophila wMelPop, but a very
particular wMelPop variant: wMelPop-PGYP (Woolfit et al. (2013) Genome Biol. Evol.).
This variant differs from the Drosophila wMelPop is several ways, the most striking
being the deletion of the Octomom genomic region, responsible for wMelPop virulence
in Drosophila (Chrostek and Teixeira (2015) PLoS Biol.). This region, present in one
copy in wMel and wMelCS, is amplified in wMelPop, while wMelPop-PGYP has zero copies
of this region (Woolfit et al. (2013) Genome Biol. Evol.). 

Lines 280-281 – wMelPop-CLA, the variant passaged through mosquito cell lines, which
was introduced into the Aedes aegypti mosquitoes (from mosquito cell lines), already
replicates much less than wMelPop in Drosophila melanogaster (McMeniman et al.
(2008). Appl. Environ. Microbiol.). It is also a null for Octomom and seems like a
more fair comparison for the wMelPop-PGYP assayed here. 

Line 300 – I would replace “fecundity” with “egg production”. 

Line 112-113 – What was the age of females blood-fed for colony maintenance? wMelPop
variants causes a range of pathologies late in life, but young insects seem to be
normal. 

Figure 4 - The loss of wMelPop from populations crossed to uninfected males seems
very fast. Could 2% of transmission failure and ~20% fewer eggs explain an almost
complete Wolbachia loss over 4 generations? More detailed method description would
help to understand this result – eg. what was the age of the females when they were
crossed to uninfected males? Were the eggs stored before hatching to provide
advantage to the uninfected individuals? Does the model described in lines 461-470
predict the loss of infection under these circumstances?

Reviewer #2: Line 128 Improper italics on Alb. Also line 130. Check throughout
manuscript. 

Line 132-138 Belongs in the results not methods. Reader will be confused/skeptical
until they get the full explanation. 

The methods are generally appropriate although the design is not ideal for comparing
fitness and other measures across time under different circumstances, etc. The
authors note this on line 275. Regardless, the data are worth examining with a grain
of salt. 

The authors are experts at all of the insect measures taken here (hatch rates, CI,
fitness, etc) and they have used the appropriate statistical analysis.

Reviewer #3: The objectives are clearly articulated, with clear testable hypotheses.
There are no concerns about statistics or ethical requirements. 

The paper would have been strengthened by the addition of a wMelPop line from a
different laboratory, although this may not have been possible. The generality of
the conclusions is questionable given that it is only one line/ one colony
effectively. Although the experiments are well designed to maximise sample size
within that one colony, it is still one colony. The colony was also collected from
the field, which may have imposed very strong selection on the original stock. It
would have been interesting to compare with a colony that has not left the lab. 

Specific comments: 

Line 109: by North Queensland material, do the authors mean Wolbachia-uninfected
material? Presumably but please specify as North Queensland material can also mean
Wolbachia infected. 

Methods – please also provide detail on rearing of larvae

Line 120 – what does this mean? Over how many years? Which experiments were done in
which years? 

Line 128 – please give primers for the wAlb

Line 151 – it would be easier for the reader if the authors denoted wMelPop with +
and wMelPop negative populations with a minus

**Results**

-Does the analysis presented match the analysis plan?

-Are the results clearly and completely presented?

-Are the figures (Tables, Images) of sufficient quality for clarity?

Reviewer #1: The results are clearly presented, I have just a single suggestion in
this department:

Fig. 2C – “incompatible females” on the axis is misleading. How about “Females with 0
viable progeny” or “infertile females”? No CI-defining pathologies were scored here,
and the incompatibility of these females is a hypothesis.

Reviewer #2: On line 271 are these females or a mix of M and F? Hard to know how this
compares to the next few lines. Put the data on the same footing. 

Line 288. Same as above. What does ‘severely mean’? Do you have a number to compare
to the next sentence?

Reviewer #3: The analysis is consistent, with results generally well presented and
clear. 

Line 269 – this is fine but it is confounded by the fact that wMelPop mosquitoes were
collected from the field, which itself may have resulted in perhaps lower densities
or selection in the field for mosquitoes which are better able to tolerate wMelPop
infection; it would have been useful if the authors had also been able to access the
original laboratory colony?; at the very least the field origin should be discussed 

Line 319 – what is meant by population effect here? What exactly is the population? 

Line 323 – could the authors clarify if experimenter is person that the mosquitoes
are feeding on? 

Figure 2: I cannot see the data for uninfected females (2019) in A and B panels - why
is this missing?

**Conclusions**

-Are the conclusions supported by the data presented?

-Are the limitations of analysis clearly described?

-Do the authors discuss how these data can be helpful to advance our understanding of
the topic under study?

-Is public health relevance addressed?

Reviewer #1: Overall, the conclusions are supported by the data, and the challenges
of comparing them with the data from other authors obtained ten years ago are
commented on. Minor comments are listed below. 

Line 426 – limited, rather than “no” attenuation. Lifespan shortening phenotype,
differences in feeding behavior, and number of eggs per female seem to have changed
since 2009, improving the outcomes for infected colonies. 

Lines 436-438 – Duplication in Octomom is easy to reverse, but it is not the cause of
virulence of wMelPop-PGYP1. 

Lines 493-494 – or heads or legs?

The last paragraph contains a mix of vector control ideas, including CI-based
suppression and population replacement and subsequent crashing. To make this
speculation more complete, assessment of the current susceptibility of wMelPop-PGYP
to viruses could be proposed. Also, it is difficult to miss the fact that if wMelPop
variants are already difficult to maintain in laboratory mosquito colonies, they are
likely to be difficult to deploy in the field.

Reviewer #2: Line 501 What do you mean? “deleterious effects and monitoring issues” 

Line 505 Explain why "isolated areas"? Do you mean because there won't be invasion
from the outside by Wolbachia free? I think to discuss this properly you need rehash
the failed releases a bit - what did the failures look like. Possibly better in the
discussion.

Reviewer #3: Yes, the conclusions are generally supported by the data presented. 

The public health relevance is somewhat tangential. 

Comments:

Discussion at lines 472-475: mating-based transmission would surely lead to
significant rates of false positive identification only if there was a large
proportion of wMelPop uninfected females already in the population, therefore the
underlying reason for loss must lie elsewhere.

Discussion at line 482: this appears to be a phenomenon restricted to wMelPop and
wMel in this particular study, not wAlbB; has this been observed in Drosophila lines
infected with wMelPop?; it is an important observation so would it be feasible for
the authors to perform additional experiments with Drosophila flies infected with
wMel and wMelPop? 

Further point on the Wolbachia mating transmission experiments: females that have
wMePop transferred via seminal fluid would be weakly positive in qPCR of whole
bodies, as the abdomen only qPCRs are close to Cq of 29 (line 394). Generally values
close to 30 in qPCR should be suspect (for any pathogen) and require repetition.
Therefore, it should be possible to screen out weakly positive females from a
laboratory line. 

Line 493: unless adults are held singly post hatching from the field it seems
somewhat difficult to implement this recommendation of only screening unmated
adults

**Editorial and Data Presentation Modifications?**

Reviewer #1: I recommend minor revisions and suggestions for modifications are listed
in the sections above. The most important change required for publication is
clarification which Wolbachia strain was under investigation here.

Reviewer #2: (No Response)

Reviewer #3: No editorial suggestions, as the writing is very clear and the figures
are fine.

**Summary and General Comments**

Reviewer #1: The manuscript “Persistent deleterious effects of an unstable
deleterious Wolbachia infection” is a thorough analysis of the phenotypes of a
pathogenic Wolbachia strain transferred from Drosophila melanogaster to mosquito
cell lines to Aedes aegyptii over ten years ago. Important life history traits have
been measured and compared to historical data. This approach detected only a small
attenuation of deleterious Wolbachia phenotypes over time.

Reviewer #2: This is an interesting study and that few if any other authors could do,
since almost no labs have this strain. The authors also have access to some
historical data for comparison. Their findings may help revive the use of wMelPop
that has largely been dismissed as unworkable for the field. There are large
caveats, however, associated with comparing fitness measures years apart. 

Wolbachia transmission by males transmission is a novel contribution to the field,
that other researchers may wish to look out for. 

I am not entirely convinced of the author's premise that rearing in the laboratory
should have selected for less virulent Wolbachia. Life shortening does not kick in
until late in life. Most labs, knowing the line is sensitive, rear it carefully.
That means taking only the first or second gonotrophic cycle (well before aging or
fitness effects kick in) and and not leaving eggs dried down for too long. I think
the authors need to give some room for this explanation as to why the virulence has
not changed. 

Additionally, to make this more than an intellectual exercise, the authors need to do
a better job of explaining how, despite failure to spread in Vietnam and Australia,
that the wMelPop strain could still be used in the field. Expand/explain. Also in
the discussion. And what about temperature (26 in the lab vs much hotter in the
field). This strain may be particularly affected. 

The writing is largely clear. Just some areas that need to be expanded for the
uninitiated or better fleshed out with respect to their arguments.

Reviewer #3: In general the paper is well written and the studies are adequate. I
have two principal concerns as discussed above:

1) we are seeing patterns largely restricted to one colony; 

2) one of the most interesting interesting findings - of male to female transmission
of wMel and wMelPop during mating - isn't explored further through additional
experiments; is this specific to Aedes aegypti or does it occur in Drosophila? what
happens to Wolbachia in the spermathecae? there is no dissection of spermathecae to
look at viability or what happens to the bacteria over time post mating. 

I should note that wMelPop deleterious effects are also observed after many, many
generations in Drosophila. This isn't really discussed anywhere.

The title suggests wMelPop is unstable - yet it seems very stable in general over the
decade being investigated, except for one particular instance when the infection was
lost. Therefore it is unclear what the authors mean by this.

PLOS authors have the option to publish the peer review history of their article
(what does this mean?). If published, this will
include your full peer review and any attached files.

If you choose “no”, your identity will remain anonymous but your review may still be
made public.

**Do you want your identity to be public for this peer review?** For
information about this choice, including consent withdrawal, please see our
Privacy Policy.

Reviewer #1: No

Reviewer #2: No

Reviewer #3: No
---

## [Decision Letter · Decision Letter 1]

9 Mar 2020

Dear Dr. Ross,

We are pleased to inform you that your manuscript 'Persistent deleterious effects of
a deleterious Wolbachia infection' has been provisionally accepted for publication
in PLOS Neglected Tropical Diseases.

Best regards,

Sassan Asgari

Guest Editor

Robert Reiner

Deputy Editor

Reviewer's Responses to Questions

**Key Review Criteria Required for Acceptance?**

**Methods**

-Are the objectives of the study clearly articulated with a clear testable hypothesis
stated?

-Is the study design appropriate to address the stated objectives?

-Is the population clearly described and appropriate for the hypothesis being
tested?

-Is the sample size sufficient to ensure adequate power to address the hypothesis
being tested?

-Were correct statistical analysis used to support conclusions?

-Are there concerns about ethical or regulatory requirements being met?

Reviewer #1: (No Response)

Reviewer #2: (No Response)

Reviewer #3: The methods are adequately described.

**Results**

-Does the analysis presented match the analysis plan?

-Are the results clearly and completely presented?

-Are the figures (Tables, Images) of sufficient quality for clarity?

Reviewer #1: (No Response)

Reviewer #2: (No Response)

Reviewer #3: The Results are adequately described.

**Conclusions**

-Are the conclusions supported by the data presented?

-Are the limitations of analysis clearly described?

-Do the authors discuss how these data can be helpful to advance our understanding of
the topic under study?

-Is public health relevance addressed?

Reviewer #1: (No Response)

Reviewer #2: (No Response)

Reviewer #3: Yes

**Editorial and Data Presentation Modifications?**

Reviewer #1: (No Response)

Reviewer #2: (No Response)

Reviewer #3: Yes

**Summary and General Comments**

Reviewer #1: This paper describes a very nice piece of research and the authors have
addressed all of my comments.

Reviewer #2: The authors have addressed the concerns.

Reviewer #3: The authors have addressed many of the critical points of the paper.

PLOS authors have the option to publish the peer review history of their article
(what does this mean?). If published, this will
include your full peer review and any attached files.

If you choose “no”, your identity will remain anonymous but your review may still be
made public.

**Do you want your identity to be public for this peer review?** For
information about this choice, including consent withdrawal, please see our
Privacy Policy.

Reviewer #1: Yes: Ewa Chrostek

Reviewer #2: No

Reviewer #3: No

---

## [Editor Report · Acceptance letter]

26 Mar 2020

Dear Dr. Ross,

We are delighted to inform you that your manuscript, "Persistent deleterious effects
of a deleterious Wolbachia infection," has been formally accepted for publication in
PLOS Neglected Tropical Diseases.

Best regards,

Serap Aksoy

Editor-in-Chief

Shaden Kamhawi

Editor-in-Chief
